# Qualitative study exploring the design of a patient-reported symptom-based risk stratification system for suspected head and neck cancer referrals: protocol for work packages 1 and 2 within the EVEREST-HN programme

Abigail Albutt [1], John Hardman [2], Lynn McVey [3,4], Chinasa Odo [3,4], Vinidh Paleri [5], Jo Patterson [6], Sarah Webb,[5] Nikki Rousseau [1], Ian Kellar [7], Rebecca Randell [3,4]

For numbered affiliations see end of article.

**Correspondence to**
Professor Rebecca Randell;
r.randell@bradford.ac.uk

## ABSTRACT

**Introduction** Between 2009/2010 and 2019/2020, England witnessed an increase in suspected head and neck cancer (sHNC) referrals from 140 to 404 patients per 100 000 population. 1 in 10 patients are not seen within the 2-week target, contributing to patient anxiety. We will develop a pathway for sHNC referrals, based on the Head and Neck Cancer Risk Calculator. The evolution of a patient-reported symptom-based risk stratification system to redesign the sHNC referral pathway (EVEREST-HN) Programme comprises six work packages (WPs). This protocol describes WP1 and WP2. WP1 will obtain an understanding of language to optimise the SYmptom iNput Clinical (SYNC) system patient-reported symptom questionnaire for sHNC referrals and outline requirements for the SYNC system. WP2 will codesign key elements of the SYNC system, including the SYNC Questionnaire, and accompanying behaviour change materials.

**Methods and analysis** WP1 will be conducted at three acute National Health Service (NHS) trusts with variation in service delivery models and ensuring a broad mixture of social, economic and cultural backgrounds of participants. Up to 150 patients with sHNC (n=50 per site) and 15 clinicians (n=5 per site) will be recruited. WP1 will use qualitative methods including interviews, observation and recordings of consultations. Rapid qualitative analysis and inductive thematic analysis will be used to analyse the data. WP2 will recruit lay patient representatives to participate in online focus groups (n=8 per focus group), think-aloud technique and experience-based codesign and will be analysed using qualitative and quantitative approaches.

**Ethics and dissemination** The committee for clinical research at The Royal Marsden, a research ethics committee and the Health Research Authority approved this protocol. All participants will give informed consent. Ethical issues of working with patients on an urgent cancer diagnostic pathway have been considered. Findings will be disseminated via journal publications, conference presentations and public engagement activities.

## STRENGTHS AND LIMITATIONS OF THIS STUDY

⇒ This study forms part of a large multicentre study (the evolution of a patient-reported symptom-based risk stratification system to redesign the suspected head and neck cancer (sHNC) referral pathway programme) that will provide evidence about whether the implementation of a patient-reported symptom-based risk stratification system for sHNC referrals is safe, improves patient experience, leads to faster diagnosis and optimises healthcare resource use.

⇒ A scoping review of the literature and data from two PhD theses were used to generate an initial list of head and neck cancer (HNC) symptoms.

⇒ The study has substantial patient, public and other stakeholder involvement throughout all phases including patients with lived experience of HNC, members of the public without prior experience of the urgent HNC pathway and hospital staff involved in the HNC diagnostic pathway.

⇒ A key strength of the study is the use of multiple methods (observation, interviews, think-aloud techniques, experience-based codesign) and theory (theoretical framework of acceptability, theoretical domains framework and normalisation process theory) to inform the development of patient-reported symptom-based risk stratification system.

⇒ Our approach enables an in-depth exploration from multiple perspectives—but this means sacrificing breadth—only a small number of geographical locations are included.

## INTRODUCTION

In healthcare, the 'head and neck' region includes the nose, mouth, throat, voice box, thyroid and salivary glands. Many patients present to general practitioners with symptoms affecting these areas, including hoarse voice, throat discomfort, neck lumps, mouth

ulcers and difficulty swallowing. In some patients, these symptoms may be caused by a 'head and neck cancer' (HNC), so, they will be referred to the hospital for an urgent specialist opinion. In England in 2020, 2 28 482 patients were referred with suspected HNC (sHNC), making it the fifth largest group of suspected cancer referrals. After specialist assessment, the vast majority (95%) of these patients can be reassured of being cancer free, but about 5% will be diagnosed with cancer. Standard practice in the UK is for all sHNC referrals to be offered a face-to-face consultation as their first hospital contact. Unfortunately, partly due to capacity issues, 1 in 10 sHNC referrals are not seen within the 2-week target.[1] Patients have reported significant anxiety in this period making any delays very undesirable.[2]

The Head and Neck Cancer Risk Calculator (HaNC-RC) symptom inventory was developed for use by specialists. The tool was developed for risk assessment to aid referral of high-risk patients to urgent specialist clinics as patients are currently seen in a chronological order.[3] HaNC-RC-v2 was deployed as a national service evaluation of sHNC referrals undergoing remote triage in secondary care during the initial peak of the coronavirus disease 2019 pandemic.[4] Clinicians were instructed to consider both the clinical history and the outcome from HaNC-RC-v2 to decide on the management plan. Clinician-led telephone triage of 4568 referrals was recorded: 53.2% were assessed urgently, 16.4% were discharged directly and 68.7% were classed as low-risk using HaNC-RC-v2. Cancer was reported in 5.6% (n=254/4,557) and late cancers were reported in 0.4% despite urgent assessment. This work demonstrated clinicians could perform symptom-based remote triage, supported by risk stratification, medium-term outcomes were acceptable and even patients undergoing urgent assessment could have missed cancers.[4] The rapidity of progression of HNC is demonstrated by the impact delays to treatment have on outcomes. It is estimated a 1-month delay to treatment led to 437, 514 and 1415 life years lost for oropharynx, larynx and oral cavity cancers, respectively.[5] With increasing numbers of referrals, an improved pathway that identifies people at higher risk for rapid assessment could offer significant benefits.[6]

The evolution of a patient-reported symptom-based risk stratification system to redesign the sHNC referral pathway (EVEREST-HN) programme aims to develop and evaluate a patient-reported symptom-based risk stratification system as part of a new pathway for sHNC referrals (see figure 1 for a study flowchart of standard care and EVEREST-HN pathways). Patients (with help from family/carers where needed) will be asked to complete an electronic questionnaire about their symptoms soon after referral. Building on our previous work, we will develop a risk stratification system for HNC based on thousands of new referrals. Hospital staff will review the information from the symptom questionnaire and the patient's individualised risk score to advise the most appropriate management before the patient comes to hospital. Obtaining symptom information from patients before they attend

hospital, rather than waiting until their urgent specialist appointment, allows for earlier risk stratification. Using this system, higher-risk patients may have targeted investigations arranged directly, before being seen in clinic, and lower-risk patients may avoid unnecessary investigations before being reassured.

Patients vary in their confidence in describing their symptoms[7] and so, when taking the patient history, clinicians use accessible and easily interpreted questions. Therefore, it is important the questions used to elicit patient-reported symptoms are understandable to patients. Development of the EVEREST-HN pathway must be based on a comprehensive understanding of existing HNC diagnostic pathways and what patients and clinicians value. Work package 1 (WP1) aims to understand: how clinicians ask questions and decide subsequent steps for patients referred with sHNC; the language patients and clinicians use to describe symptoms[8]; how clinicians reassure and discharge low-risk patients; and patients' and clinicians' views of the current diagnostic process for HNC. WP2 aims to codesign key elements of the SYmptom iNput Clinical (SYNC) system, including the SYNC Symptom Questionnaire and behaviour change intervention materials to integrate the system into existing hospital workflows. Table 1 describes the subsequent WPs (3–6) of the EVEREST-HN programme and demonstrates how WP1 and WP2 integrate into the wider programme of work. Figure 2 shows an overview of how the WPs interlink.

## METHODS AND ANALYSIS
### WP1: qualitative research to generate SYNC Symptom Questionnaire
#### Study design
WP1 will employ qualitative methods including interviews, observation and recordings of consultations and analysis of referral documentation. Work will be conducted at three sites with variation in service delivery models, including those not traditionally involved in research. Sites will be recruited to ensure a broad mixture of social, economic and cultural backgrounds of potential participants. Approximately 150 adults (n=50 per site) referred for sHNC and 15 clinicians (n=5 per site including different subspecialties that see patients with sHNC) will be recruited. A subset of recruited patients (approximately n=30) will be interviewed. Clinicians (n=15 varying grades and specialty) will be interviewed and interviews may include a review of extracts of recordings of their consultations to understand actions taken within and as a result of the consultation. Data collection started in April 2023 and is due to be completed in April 2024.

#### Sample
Sites will be chosen for variation in service delivery model and to facilitate diversity in patient characteristics. Patient recruitment will aim for variation in age,

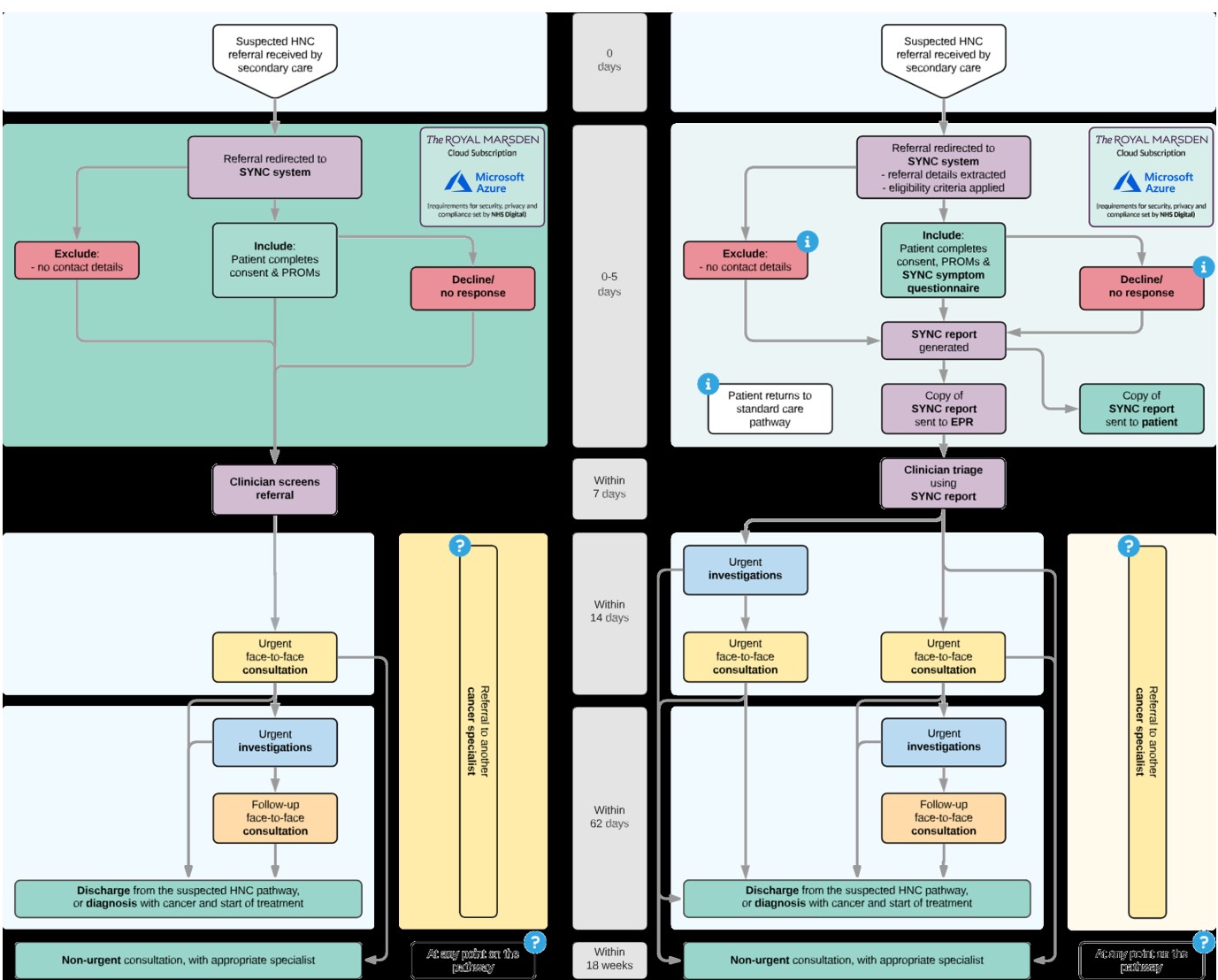

**Figure 1** Study flowchart showing standard care and evolution of a patient-reported symptom-based risk stratification system to redesign the suspected head and neck (HNC) referral pathways. SYNC, SYmptom iNput Clinical. PROMs, patient-reported outcome measures. EPR, electronic patient record.

ethnicity, socioeconomic status and presenting symptoms. Clinicians recruited will be clinical staff involved in HNC diagnosis, varying in seniority (consultant, registrar) and specialty (ear, nose and throat surgery, oral and maxillofacial surgery). Recruitment is not restricted to English speakers. Where a consultation is conducted with the support of an interpretation service or is conducted in a language other than English (where recruited clinicians speak the language of the patient), these patients are eligible for inclusion in the study. Study materials are available in the most common non-English languages at each site to facilitate this.

Our target sample sizes have been selected to balance breadth and depth. We aim not only to achieve variety in terms of participant characteristics but also an in-depth understanding of how clinicians vary in their approach to the diagnostic consultation. Our target sample size of 150 has been chosen to enable sufficient patients per clinician (approximately 10 patients per clinician) to identify

variation in approach to the diagnostic consultation and also variation in clinical (particularly HaNC-RC criteria) and sociodemographic characteristics of patients at each site. An initial phase of recruitment will involve recordings of one or two clinics per participating clinician at each site (n=approximately 7 consenting patients per clinician, total n=105). Subsequent recruitment (estimate additional n=45) will target patients referred with symptoms or characteristics not well represented in the initial sample (referral letters will be reviewed by hospital staff to identify potential patients). Recruitment will be monitored frequently throughout data collection/analysis to ensure that the range of characteristics and numbers of participants are adequate to enable our research questions to be addressed. Recruitment will cease when we have 'adequate data to tell a rich, complex and multifaceted storey about patternings related to the phenomena of interest'.[9] The aim of our rapid qualitative analysis is to provide an understanding of the language used and

**Table 1** Brief description of EVEREST-HN programme work packages (WPs) 3–6

| WPs | Brief description |
|---|---|
| 3: Informatics linkage of national datasets for suspected HNC referrals | To use data from routinely collected electronic health records to describe the sHNC population and their diagnostic pathway and provide prognostic information for WP4 and WP6. |
| 4: SYNC system development phase 3: Feasibility study to finalise EVEREST-HN pathway | To conduct a concurrent data collection exercise and feasibility study, using the SYNC Symptom Questionnaire developed in WP1 and WP2, alongside the prognostic data from WP3, to develop the SYNC risk stratification algorithm, agree management thresholds and recommendations by stakeholder consensus and finalise all elements of the EVEREST-HN pathway. |
| 5: Cluster randomised controlled trial | To conduct a cluster randomised trial, with 6-month pilot, to compare the new pathway to standard care, using a primary outcome of cancer diagnosed within 62 days. |
| 6: Health economics | To evaluate the cost-effectiveness of the pathway. |

EVEREST-HN, evolution of a patient-reported symptom-based risk stratification system to redesign the suspected head and neck cancer referral pathway; HNC, head and neck cancer; sHNC, suspected HNC; SYNC, SYmptom iNput Clinical.

key requirements for the SYNC system, not necessarily to generate themes or a theory. We have previously referred to this point in rapid qualitative analysis as 'analytic saturation'.[10]

### Inclusion criteria
Patients: adults≥18 years referred via sHNC pathway without previous history of HNC. Participant is willing

and able to give informed consent for participation in the study.

Clinicians: Staff at participating sites involved in the diagnostic pathway for people with sHNC.

### Exclusion criteria
Individuals who do not meet the inclusion criteria.

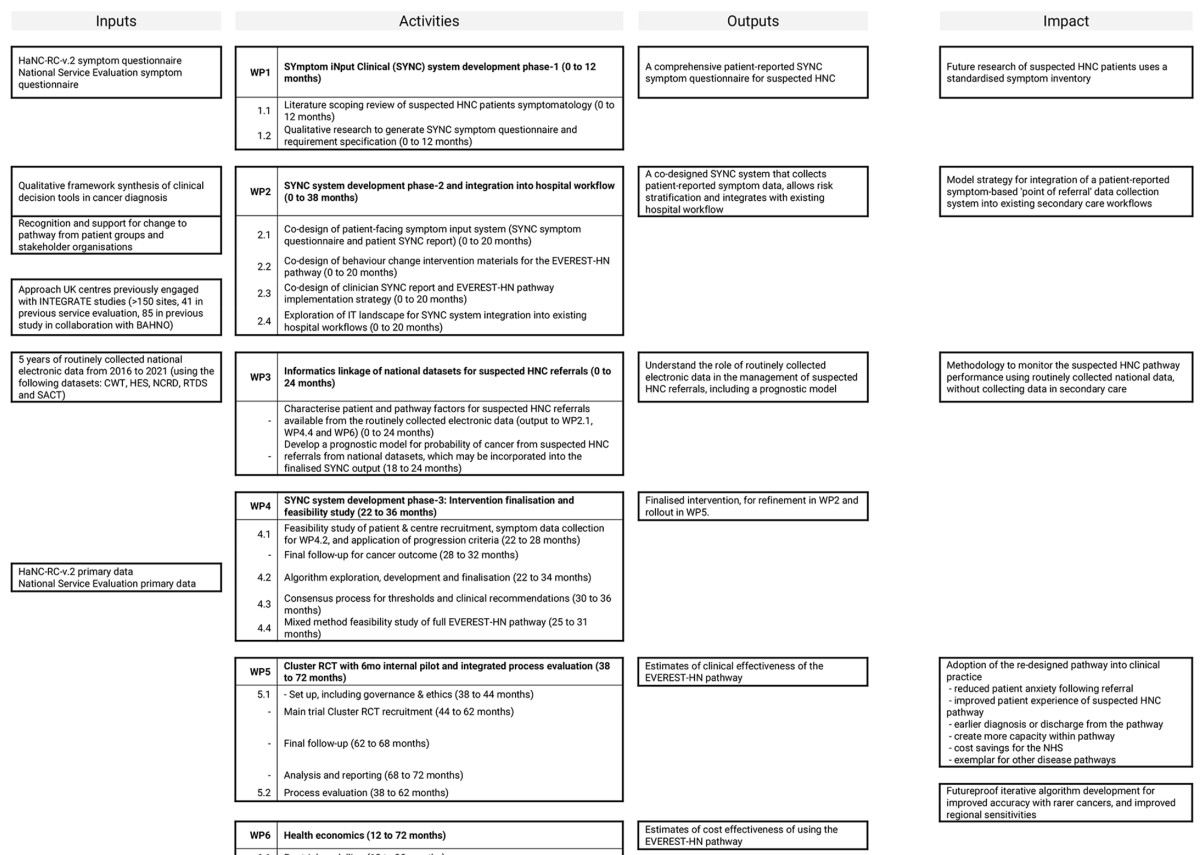

**Figure 2** Overview of evolution of a patient-reported symptom-based risk stratification system to redesign the suspected head and neck cancer referral pathway (EVEREST-HN) work packages.

## Methodology

### Consent

Members of the local clinical team will proactively identify eligible patients through triage of referral letters. Eligible patients will be those allocated to attend a diagnostic clinic or with symptom(s) or other characteristics identified as being important to include in the sample. Where possible, the patient information sheet will be sent to patients with appointment details.

Participants will be approached and consented by local care teams, who will also confirm eligibility. Written consent for consultations to be recorded will be taken immediately prior to appointments. Where patients are willing, consent will also be taken for their contact details to be shared with the research team, so they can potentially be invited to participate in an interview. Verbal consent will be taken and recorded prior to interviews conducted remotely.

The EVEREST-HN qualitative researchers will conduct a small number of clinic observations and qualitative research with participants who have consented to be involved. For clinic observations, informed consent will be received for both clinician and patient participants.

### Data collection

Data collection will comprise:
► Non-participant observation and audio recordings of diagnostic consultations.
► Review of referral documentation.
► Patient interviews: the focus will be on their experience of the current diagnostic pathway; we will also seek their views on the EVEREST-HN pathway.
► Clinician interviews: the focus will be on the clinical decision-making process, including views on the EVEREST-HN pathway.

Our recruitment strategy has been codeveloped with patient and public involvement (PPI) and with reference to NIHR INCLUDE guidance.[11] Recruitment information will be available in different formats and languages. Topic guides for the interviews will be informed by our own and other research, our PPI and by our theoretical framework including theoretical framework of acceptability (TFA)[12] and normalisation process theory (NPT)[13] which can help to sensitise researchers to relevant aspects of context.[14] See online supplemental appendices 1 and 2 for patient and clinician interview topic guides, respectively.

### Dataset and case report form (CRF)

Limited descriptive information will be captured to monitor included participants' characteristics, consider subgroups in the qualitative analysis and describe participants in study outputs.

The following fields will be recorded on the CRF:
► Demographics (age, gender, ethnicity).
► Referral criteria (eg, throat lump—based on standard referral forms).
► Smoking status.

► Outcome of initial consultation (discharge, further tests, referral to other service).
► Diagnostic outcome (HNC diagnosed, other/no cancer diagnosis).

### Pseudonymisation

Sites will generate a unique study ID for each participant and use a 'key' to reference this to the NHS and hospital medical record number. This key will be stored locally at contributing trusts on an Excel file on the hard drive of a secure NHS computer. The study key will be stored for the duration of the study and then destroyed in line with local processes for handling patient identifiable data.

Study ID will comprise a single letter hospital code followed by a three-digit consecutive number, for example, A001, A002, etc.

### Theoretical framework

Data collection and analysis will be informed by our theoretical framework which includes TFA and NPT.[12 13] TFA consists of seven constructs: affective attitude; burden; ethicality; intervention coherence; opportunity costs; perceived effectiveness; and self-efficacy.[12] It can be used to highlight characteristics of healthcare interventions that could be improved. NPT considers factors that affect implementation in four areas: coherence; cognitive participation; collective action; and reflexive monitoring.[13] NPT has been widely used in studies on implementation of interventions in healthcare including in trial process evaluations,[15 16] e-health[17] and decision aids.[18 19] In e-health, its use has highlighted the importance of sense-making (users' understanding of the e-health intervention's purpose and its differences from usual practice), the impact of digital interventions on roles and responsibilities and the work that must be done to incorporate e-health interventions into pathways.[17 20] Our theoretical framework will be reviewed regularly throughout WP1 and WP2. We will consider whether the existing framework fits the data collected and whether additional theories, such as the non-adoption, abandonment, scale-up, spread and sustainability framework,[21] would be helpful.

### Data storage

#### Study database

Descriptive information about patients and their health will be entered by site staff into MACRO, a secure database, for transfer to and storage at the University of Leeds. Participants will be identified by their unique study ID.

Qualitative data will be translated to English (where required) and transcribed by an approved third party before being anonymised. The original recordings will be stored securely separately from transcripts and retained for a minimum of 5 years after project end for data verification, if required. Contact information will be stored until it is no longer needed to contact participants.

Anonymised transcripts and associated pseudonymised descriptive information about participants will be retained after the end of the study to enable data sharing.

It will be possible to link the transcript to the descriptive information via the study ID, as this is likely to be essential information as part of data sharing, but after the contact information has been deleted it will not be possible for anyone to link the anonymised information to any identifiable information.

## Data analysis

All interviews will be audio recorded and transcribed verbatim. Both transcripts and contemporaneous field notes from non-participant observation in clinical settings will be edited to ensure participant anonymity. Analysis will proceed concurrently with data collection using a rapid qualitative analysis approach and will be conducted according to standard procedures of rigorous rapid qualitative analysis.[22 23] Separate rapid assessment procedure (RAP) sheets[22] will be developed for patient interviews, clinician interviews and consultations based on the research questions. Researchers will independently complete RAP sheets after each interview and for each recorded consultation. We will hold 'data clinics' where the research team exchanges interpretations of key issues emerging from the data. These will facilitate decisions regarding analytic saturation.[10 24]

Content analysis of consultation recordings will focus on language used, generating a glossary of terms used by patients to describe their symptoms and aiming to identify language used by clinicians which is helpful in generating relevant responses from patients, as well as terms that seem to be poorly understood. Thematic analysis of interviews will inform guiding principles (intervention objectives and key design features) and be used to develop a requirement specification and 'personas' for codesign. The requirement specification will detail both functional requirements (what the SYNC system should do) and non-functional requirements (including look and feel, usability, performance and maintainability and support requirements and requirements relating to implementation).[25] The requirements will be presented using the Volere format.[26]

## WP2: codesign of the SYNC Symptom Questionnaire
### Study design

The codesign process will be iterative and will involve a series of interactive focus groups, think-aloud technique[27] and experience-based codesign (EBCD).[28]

### Sample

An ethnic diverse group including participants with English as a second language will be recruited from PPI groups at the University of Bradford and their networks. We will also involve the EVEREST-HN PPI representatives, to provide their experience of the HNC pathway. Based on previous experience of codesign, we aim to have eight participants.

## Methodology
### Data collection and analysis
*Focus groups*

These will be undertaken via Microsoft Teams and recorded. Each focus group will last approximately 90 min. Codesign requires not only careful preparation but also flexibility, to allow participant involvement in the design of the approach and the intervention.[29] While open to revision and further specification, we will move from discussing personas and gathering participants' perspectives on what the questionnaire should look like to discussion of a paper-based prototype to encourage further design input and discussion of a high fidelity prototype, in terms of how comprehensible, usable and acceptable it is. Throughout the focus groups the moderator will encourage discussion of differences of opinions.[30] At the end of each focus group, the moderator will present any tentatively identified issues to participants for confirmation or clarification.[31] Participants will receive incentives for their time and contribution towards the codesign process.

Audio recordings will be transcribed verbatim. Data analysis will be conducted after each focus group using thematic analysis and will contribute towards refinements to the requirement specification and determine priorities for subsequent prototypes. At each stage, the requirement specification and emerging design ideas will be shared and discussed with the SYNC system developers. Additionally, transcripts will be analysed using the TFA[12] and the theoretical domains framework/mechanisms of action (TDF/MOA).[32–34] Where this suggests a need for behavioural support, the findings will carry forward to the EBCD.

### Think-aloud technique

An ethnically diverse group of participants who are naive to codesign focus groups and vary in confidence in using computers will be asked to think aloud as they complete tasks designed to enable them to explore key prototype functionality, observed by the researcher. Audio recording will capture participants' comments as they conduct the tasks. Following completion of tasks, participants will be asked to complete the System Usability Scale (SUS), a flexible questionnaire designed to assess any technology.[35] The SUS is quick and easy to complete, consisting of 10 statements scored on a 5-point scale, with final scores ranging from 0 to 100 (with a SUS Score above a 68 being considered above average). Audio recordings of the codesign focus groups will be transcribed verbatim and thematically analysed. Where this suggests changes to the system, these will be the basis of a prioritised list of revisions to be made. Descriptive statistics will be produced for the SUS Score.

### Experience-based codesign

A behavioural support intervention for patients will be developed using an EBCD approach. We will produce an animated catalyst film to stimulate codesign activities and

will follow the EBCD process of showing the film to staff involved in the sHNC diagnosis pathway and patients, first separately and then jointly. The catalyst film will make use of personas and themes from the focus groups and participant quotes that exemplified themes found in the TDF/MOA informed deductive thematic analysis of focus group recordings. This is more robust than standard EBCD, in that thematic analysis is applied, rather than the informal editing process specified in EBCD. We will undertake stakeholder prioritisation exercises in the workshops, using the personas to undertake user journey mapping and pain point analysis and then using prioritisation/ranking activities to agree on areas of focus for behavioural support intervention development. Based on focus group findings, the EVEREST-HN study team will identify prior behavioural support materials used in usual care in sHNC referral pathways. We will code the content of these materials using the behaviour change technique (BCT) taxonomy,[34 36] and label them according to the links to TDF/MOA constructs. Together, the catalyst film and the prior behavioural support content will provide material from which the stakeholder codesign team can together produce the novel behavioural support intervention for the pathway for sHNC referrals to secondary care, employing ideation techniques, sketching and prototyping methods.[37]

Up to four workshops with patients and staff will explore solutions to the barriers and facilitators to acceptability and engagement with the codesigned pathway, using findings from the TFA/TDF/MOA-informed deductive thematic analysis represented in the catalyst film. Coproduction activities will focus on adapting existing materials to bolster their acceptability, focusing first on the constructs, then ranking the identified prior materials for each construct on TFA and finally adapting prior materials on these dimensions to improve their application to the pathway and acceptability to patients. Where suitable materials do not exist, these will be developed using BCT definitions linked to the identified TDF/MOA constructs as a starting point, and codesign activities will be structured with a focus on enhancing TFA constructs.

The materials will be coded for their BCT content and linked to the TDF/MOA framework by a member of the study team not involved in codesign activities. This will form the basis of a TFA/TDF/MOA-informed comparative deductive thematic analysis of the coproduction activities of the workshops and the content of the intervention materials. We will operationalise findings relating to the BCT content of the finalised materials and their linked TDF/MOA constructs using the Grid-Enabled Measures database[38] to identify related scales for adaptation by the study team. These will index the effects of the behavioural support intervention to support engagement with the pathway.

## Patient and public involvement
We worked with three PPI groups to define the aims, objectives and design of the EVEREST-HN programme.

Patients felt strongly that any new system should be as safe at detecting cancer as the current one. Patients prioritised being seen as quickly as possible when referred through the 2-week wait system. Additionally, they felt that the period between referral and completing the pathway (either reassurance/discharge or diagnosis with cancer) was a very anxious time, describing a 'feeling of limbo', of being in a 'vacuum' and that the 'worst part of it was the waiting'. The groups reported earlier contact, even if delivered remotely, would reduce anxiety. We have enlisted two PPI representatives to ensure that the patient perspective remains core to the EVEREST-HN programme. Both are HNC survivors. They will be active members of the project management group, contributing to the running of the project, being involved in decision-making, collaboratively pre-empting barriers, highlighting any delays against the proposed timeline and jointly generating solutions and providing their unique perspective. An ethnically diverse group of participants, including people with English as a second language, will be recruited from PPI groups at the University of Bradford and their networks to codesign the SYNC Symptom Questionnaire.

## Ethics and dissemination
This protocol has been reviewed and approved by the committee for clinical research at The Royal Marsden (reference: CCR5683), a research ethics committee and the Health Research Authority (REC reference: 22/NW/0327). All participants will give fully informed consent to take part in the study. Consideration of the ethical issues of working with patients on an urgent cancer diagnostic pathway has been a priority in the design of our data collection.

Our study design has been informed by our existing work with patients with HNC, including with patients shortly after diagnosis. We have found that, if the reasons for approaching people at a particular time are made clear and the approach is made with sensitivity, patients appreciate the need for them to contacted at that time, are willing to take part and may see participation as an opportunity for something positive to come out of a difficult situation. Our PPI representatives will continue to provide a patient perspective throughout study design and conduct. Specifically, we will work with them in operationalising our recruitment and data collection approaches.

Our PPI representatives suggested that the time prior to diagnosis is an acceptable time to conduct research for WP1, because at this point waiting can be very stressful, and having the opportunity to talk about the process is unlikely to add additional anxiety. The period around diagnosis is likely to be more problematic; we wish to avoid contacting people in the period immediately following diagnosis. However, the period following this, early in the treatment process, before side effects become too challenging, is also likely to be an acceptable time to conduct qualitative interviews with patients. Any recruitment after

the initial diagnostic consultation will be led by sites, who will know where patients are in their diagnosis and treatment pathway. Involvement of research nurses to obtain informed consent for participation will help to avoid additional pressure that patients might feel to participate.

In terms of risks to recruitment, poor recruitment either of patients in general or patients in specific groups could lead to an inadequate dataset or data which are not reflective of the diversity of patients. Slow recruitment also risks delaying subsequent WPs. Our PPI is key to developing a recruitment strategy which is engaging and appropriate. Our experts by experience will also assist us with trouble shooting in the event of slower than anticipated recruitment. Within the EVEREST-HN programme there is planned resource to allow for involvement of research nurses and ear, nose and throat trainees at sites to support site and recruitment activity. We will use various methods to contact potential patient participants for interview including telephone and email, in line with their preferred contact method. Shared working across WP1 and WP2 will ensure timely sharing of interim findings—mitigating the risk of delays impacting onward WPs.

Findings from the study will be submitted for publication in relevant peer-reviewed journals. Reporting guidance for qualitative research will be followed. We will also work with our dissemination and impact advisory group to identify other routes to dissemination for varying stakeholder groups, including patients and public, clinicians and researchers, cancer alliances and the technology sector. This may include material in multiple formats (presentations, social media content, infographics) to target different audiences.

**Author affiliations**
[1]Leeds Institute of Clinical Trials Research, University of Leeds, Leeds, UK
[2]Northwick Park Hospital, London, UK
[3]Centre for Digital Innovations in Health & Social Care, University of Bradford, Bradford, UK
[4]Wolfson Centre for Applied Health Research, Bradford, UK
[5]The Royal Marsden NHS Foundation Trust, London, UK
[6]University of Liverpool, Liverpool, UK
[7]Department of Psychology, The University of Sheffield, Sheffield, UK

**Acknowledgements** The authors would like to thank Chris Elkington and John Holmes (experts by experience) for their contribution in conceptualisation of the study and protocol development.

**Contributors** JH, IK, JP, VP, NR and RR conceptualised the study. NR leads WP1 and RR and IK lead WP2. IK, RR and NR are joint senior authors of this manuscript. VP and JP are joint chief investigators for the EVEREST programme. JH is deputy chief investigator for the EVEREST programme. All authors contributed to protocol development. AA wrote the first draft of the manuscript, and IK, LM, CO, JP, VP, NR, RR and SW reviewed and contributed to subsequent drafts. All authors reviewed and approved the final draft of the manuscript. NR, IK and RR are joint final authors.

**Funding** This work was supported by NIHR Programme Grants for Applied Research (grant number: NIHR202862).

**Competing interests** None declared.

**Patient and public involvement** Patients and/or the public were involved in the design, or conduct, or reporting, or dissemination plans of this research. Refer to the Methods and analysis section for further details.

**Patient consent for publication** Not applicable.

**Provenance and peer review** Not commissioned; externally peer reviewed.

**ORCID iDs**
Abigail Albutt http://orcid.org/0000-0002-3524-7930
John Hardman http://orcid.org/0000-0002-6591-5119
Lynn McVey http://orcid.org/0000-0003-2009-7682
Chinasa Odo http://orcid.org/0000-0002-0770-0806
Vinidh Paleri http://orcid.org/0000-0002-7933-4585
Jo Patterson http://orcid.org/0000-0001-8898-8292
Nikki Rousseau http://orcid.org/0000-0001-8826-3515
Ian Kellar http://orcid.org/0000-0003-1608-5216
Rebecca Randell http://orcid.org/0000-0002-5856-4912

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
