## [Reviewer comments · BMJ Open]

ARTICLE DETAILS

TITLE (PROVISIONAL)	A qualitative study exploring the design of a patient-reported symptom-based risk stratification system for suspected head and neck cancer referrals: protocol for work packages 1 and 2 within the EVEREST-HN programme
AUTHORS	Albutt, Abigail; Hardman, John; McVey, Lynn; Odo, Chinasa; Paleri, Vinidh; Patterson, Jo; Webb, Sarah; Rousseau, Nikki; Kellar, Ian; Randell, Rebecca

VERSION 1 – REVIEW

REVIEWER	Wei, Xing Sichuan Cancer Hospital and Research Institute, Department of Thoracic Surgery
REVIEW RETURNED	06-Dec-2023

GENERAL COMMENTS	Dear authors: Thank you for inviting me to review this research protocol. This is a well-thought-out multipronged study, whose main goal is to provide patients with more accurate and personalized diagnoses for head and neck cancer. By integrating qualitative research methods, patient and public involvement (PPI), and theoretical frameworks, the study aims to develop a symptom-based risk stratification system, which is crucial for improving diagnostic efficiency and accuracy. This study is important for enhancing early diagnosis and treatment outcomes for head and neck cancer. By gaining a deeper understanding of patients' experiences and perceptions during the diagnostic process, the research not only helps to optimize existing diagnostic pathways but also increases patient satisfaction and engagement. Additionally, the design of the study takes into account ethical considerations and cultural diversity, ensuring fairness and inclusivity in the research. Overall, this study is meticulously designed and methodologically sound, having significant practical implications for improving the diagnostic and treatment pathways for patients with head and neck cancer. I have only a few minor suggestions for modifications for your consideration. I sincerely hope that this study will be published, as it can bring new insights to the medical community and tangible benefits to patients. This is an excellent study that deserves widespread attention and recognition. I am looking forward to seeing the results of this part of the research, as well as the subsequent studies in this series.
--

	1. One suggestion for improvement is the inclusion of a flowchart to visually represent the research process. A flowchart can significantly enhance the readability and comprehension of the study design, particularly for complex research projects like this one. Flowchart are not only beneficial for readers to grasp the study's methodology and progression quickly but also aid in highlighting the interconnections between different components of the research. This can be particularly useful in multi-phase studies, where each stage builds upon the previous one. 2. The protocol insufficiently assesses potential risks and challenges, such as difficulties in participant recruitment and data quality control. Consideration should be given to adding risk assessment and mitigation strategies. 3. Additionally, has this study been registered on the relevant clinical research website? If so, please provide the corresponding registration information.
--	---

REVIEWER	Goepfert, Ryan The University of Texas MD Anderson Cancer Center Division of Surgery
REVIEW RETURNED	24-Dec-2023

GENERAL COMMENTS	Thank you for this most interesting and thoughtful approach to a very challenging problem. I wish you luck! Please address the following minor edits:  -Define all acronyms (most are clearly defined but some even if obvious (EVEREST-HN, GP, TFA/NPT) should be defined. -Will patient eligibility in WP1 be restricted to english speakers/readers? -In WP2, is there a proposed length to Teams sessions, compensation to patients/participants, or an option for a second session/debrief?
--

VERSION 1 – AUTHOR RESPONSE

Reviewer 1	
One suggestion for improvement is the inclusion of a flowchart to visually represent the research process. A flowchart can significantly enhance the readability and comprehension of the study design, particularly for complex research projects like this one. Flowchart are not only beneficial for readers to grasp the study's methodology and progression quickly but also aid in highlighting the interconnections between different components of the research. This can be particularly useful in multi-phase studies, where each stage builds upon the previous one.	Thank you, we agree that your suggestion of a flowchart would aid comprehension of the study and wider programme. We have added one flow chart showing standard care and the EVEREST-HN pathway and one diagram showing an overview of work packages within the EVEREST-HN programme.
The protocol insufficiently assesses potential risks and challenges, such as difficulties in participant recruitment and data quality control. Consideration should be given to adding risk assessment and mitigation strategies.	Thank you for highlighting this. We have added a section about risks and challenges to the main document in the ethics and dissemination section.

Additionally, has this study been registered on the relevant clinical research website? If so, please provide the corresponding registration information.	This study has not been registered but subsequent trial work packages will be registered on a trial website.
Reviewer 2	
Define all acronyms (most are clearly defined but some even if obvious (EVEREST-HN, GP, TFA/NPT) should be defined.	Thank you for your comment. All acronyms have now been defined including EVEREST-HN, GP, TFA/NPT.
Will patient eligibility in WP1 be restricted to english speakers/readers?	Thank you for highlighting this. Patient eligibility in WP1 will not be restricted to English speakers/readers. Where a consultation is conducted with the support of interpretation services or when recruited clinicians speak the language of the patient and consultations are conducted in another language, these patients are eligible for inclusion in the study. Study materials are available in commonly used languages at each of the hospital sites. We have added this information to the sample and study database sections of the protocol.
In WP2, is there a proposed length to Teams sessions, compensation to patients/participants, or an option for a second session/debrief?	Thank you for highlighting this. This information has been added in the focus group section within 'data collection and analysis'.